# AUGMENTED SLICED WASSERSTEIN DISTANCES

## ABSTRACT

While theoretically appealing, the application of the Wasserstein distance to large-scale machine learning problems has been hampered by its prohibitive computational cost. The sliced Wasserstein distance and its variants improve the computational efficiency through random projection, yet they suffer from low projection efficiency because the majority of projections result in trivially small values. In this work, we propose a new family of distance metrics, called augmented sliced Wasserstein distances (ASWDs), constructed by first mapping samples to higher-dimensional hypersurfaces parameterized by neural networks. It is derived from a key observation that (random) linear projections of samples residing on these hypersurfaces would translate to much more flexible *nonlinear* projections in the original sample space, so they can capture complex structures of the data distribution. We show that the hypersurfaces can be optimized by gradient ascent efficiently. We provide the condition under which the ASWD is a valid metric and show that this can be obtained by an injective neural network architecture. Numerical results demonstrate that the ASWD significantly outperforms other Wasserstein variants for both synthetic and real-world problems.

## 1 INTRODUCTION

Comparing samples from two probability distributions is a fundamental problem in statistics and machine learning. The optimal transport (OT) theory (Villani, 2008) provides a powerful and flexible theoretical tool to compare degenerative distributions by accounting for the metric in the underlying spaces. The Wasserstein distance, which arises from the optimal transport theory, has become an increasingly popular choice in various machine learning domains ranging from generative models to transfer learning (Gulrajani et al., 2017; Arjovsky et al., 2017; Kolouri et al., 2019b; Lee et al., 2019; Cuturi & Doucet, 2014; Claici et al., 2018; Courty et al., 2016; Shen et al., 2018; Patrini et al., 2018).

Despite its favorable properties, such as robustness to disjoint supports and numerical stability (Arjovsky et al., 2017), the Wasserstein distance suffers from high computational complexity especially when the sample size is large. Besides, the Wasserstein distance itself is the result of an optimization problem — it is non-trivial to be integrated into an end-to-end training pipeline of deep neural networks, unless one can make the solver for the optimization problem differentiable. Recent advances in computational optimal transport methods focus on alternative OT-based metrics that are computationally efficient and differentiably solvable (Peyré & Cuturi, 2019). Entropy regularization is introduced in the Sinkhorn distance (Cuturi, 2013) and its variants (Altschuler et al., 2017; Dessein et al., 2018; Lin et al., 2019) to smooth the optimal transport problem; as a result, iterative matrix scaling algorithms can be applied to provide significantly faster solutions with improved sample complexity (Genevay et al., 2019).

An alternative approach is to approximate the Wasserstein distance through *slicing*, i.e. linearly projecting, the distributions to be compared. The sliced Wasserstein distance (SWD) (Bonneel et al., 2015) is defined as the expected value of Wasserstein distances between one-dimensional random projections of high-dimensional distributions. The SWD shares similar theoretical properties with the Wasserstein distance (Bonnotte, 2013) and is computationally efficient since the Wasserstein distance in one-dimensional space has a closed form solution based on sorting. Deshpande et al. (2019) extends the sliced Wasserstein distance to the max-sliced Wasserstein distance (max-SWD), by finding a single projection direction with the maximal distance between projected samples. In Nguyen et al. (2020), the distributional sliced Wasserstein distance (DSWD) finds a distribution of projections that maximizes the expected distances over these projections. The subspace robust Wasserstein distance extends the idea of slicing to projecting distributions on linear subspaces (Paty & Cuturi, 2019). However, the linear nature

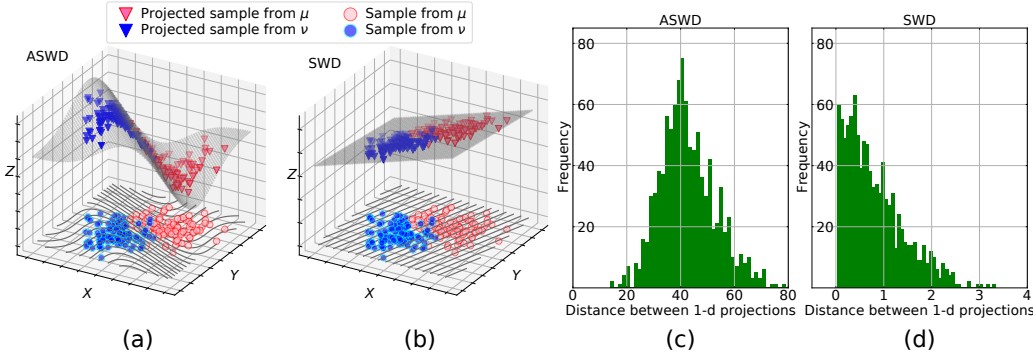

Figure 1: (a) and (b) are visualizations of projections for the ASWD and the SWD between two 2-dimensional Gaussians. (c) and (d) are distance histograms for the ASWD and the SWD between two 100-dimensional Gaussians. Figure 1(a) shows that the injective neural network embedded in the ASWD learns data patterns (in the $X$-$Y$ plane) and produces well-separate projected values ($Z$-axis) between distributions in a random projection direction. The high projection efficiency of the ASWD is evident in Figure 1(c), as almost all random projection directions in a 100-dimensional space lead to significant distances between 1-dimensional projections. In contrast, random linear mappings in the SWD often produce closer 1-d projections ($Z$-axis) (Figure 1(b)); as a result, a large percentage of random projection directions in the 100-d space result in trivially small distances (Figure 1(d)), leading to a low projection efficiency in high-dimensional spaces.

of these projections usually leads to low projection efficiency of the resulted metrics in high-dimensional spaces (Deshpande et al., 2019; Liutkus et al., 2019; Kolouri et al., 2019a).

More recently, there are growing interests and evidences that slice-based Wasserstein distances with nonlinear projections can improve the projection efficiency, leading to a reduced number of projections needed to capture the structure of the data distribution (Kolouri et al., 2019a; Nguyen et al., 2020). (Kolouri et al., 2019a) extends the connection between the sliced Wasserstein distance and the Radon transform (Abraham et al., 2017) to define generalized sliced Wasserstein distances (GSWDs) by utilizing generalized Radon transforms (GRTs). It is shown in (Kolouri et al., 2019a) that the GSWD is indeed a metric if and only if the adopted GRT is injective. Injective GRTs are also used to extend the DSWD to the distributional generalized sliced Wasserstein distance (DGSWD) (Nguyen et al., 2020). However, both the GSWD and the DGSWD are restricted by the limited class of injective GRTs, which utilize the circular functions and a finite number of harmonic polynomial functions with odd degrees as their defining function (Kuchment, 2006; Ehrenpreis, 2003). The results reported in (Kolouri et al., 2019a; Nguyen et al., 2020) show impressive performance from the GSWD and the DGSWD, yet they require one to specify a particular form of defining function from the aforementioned limited class of candidates. However, the selection of defining function is usually task-dependent and needs domain knowledge. In addition, the impact on performance from different defining functions is still unclear.

One variant of the GSWD (Kolouri et al., 2019a) is the GSWD-NN, which generates projections *directly* with neural network outputs to remove the limitations of slicing distributions with predefined GRTs. In the GSWD-NN, the number of projections, which equals the number of nodes in the neural network's output layer, is fixed. Hence different neural networks are needed if one wants to change the number of projections. There is also no random projections involved in the resulted GSWD-NN, since the projection results are determined by the neural network's weights. Besides, the GSWD-NN is a *pseudo-metric* since it uses a vanilla neural network, rather than the Radon transform or GRTs, as its push-forward operator. Therefore, the GSWD-NN does not fit into the theoretical framework of the GSWD and does not inherit its geometric properties.

In this paper, we present the augmented sliced Wasserstein distance (ASWD), a distance metric constructed by first mapping samples to hypersurfaces in an *augmented* space, which enables flexible nonlinear slicing of data distributions for improved projection efficiency (See Figure 1). Our main contributions include: (i) We exploit the capacity of nonlinear projections employed in the ASWD by constructing injective mapping with arbitrary neural networks; (ii) We prove that the ASWD is a valid distance metric; (iii) We provide a mechanism in which the hypersurface where high-dimensional distributions are projected onto can be

optimized and show that the optimization of hypersurfaces can help slice-based Wasserstein distances improve their projection efficiency. Hence, the ASWD is data-adaptive, i.e. the hypersurfaces can be learned from data. This implies one does not need to specify a defining function from a limited class of candidates; (iv) We demonstrate superior performance of the ASWD in numerical experiments for both synthetic and real-world datasets.

The remainder of the paper is organized as follows. Section 2 reviews the necessary background. We present the proposed method and its numerical implementation in Section 3. Numerical experiment results are presented and discussed in Section 4. We conclude the paper in Section 5.

## 2 BACKGROUND

In this section, we provide a brief review of concepts related to the proposed work, including the Wasserstein distance, (generalized) Radon transform and (generalized) sliced Wasserstein distances.

**Wasserstein distance:** Let $P_k(\Omega)$ be a set of Borel probability measures with finite $k$-th moment on a Polish metric space $(\Omega, d)$ (Villani, 2008). Given two probability measures $\mu, \nu \in P_k(\Omega)$, whose probability density functions (PDFs) are $p_\mu$ and $p_\nu$, the Wasserstein distance of order $k \in [1, +\infty)$ between $\mu$ and $\nu$ is defined as:

$$W_k(\mu,\nu) = \left( \inf_{\gamma \in \Gamma(\mu,\nu)} \int_\Omega d(x,y)^k d\gamma(x,y) \right)^{\frac{1}{k}}, \tag{1}$$

where $d(\cdot,\cdot)^k$ is the cost function, $\Gamma(\mu,\nu)$ represents the set of all transportation plans $\gamma$, i.e. joint distributions whose marginals are $p_\mu$ and $p_\nu$, respectively. With a slight abuse of notation, we interchangeably use $W_k(\mu,\nu)$ and $W_k(p_\mu,p_\nu)$.

While the Wasserstein distance is generally intractable for high-dimensional distributions, there are several favorable cases where the optimal transport problem can be efficiently solved. In particular, if $\mu$ and $\nu$ are continuous one-dimensional measures, the Wasserstein distance between $\mu$ and $\nu$ has a closed form solution (Bonneel et al., 2015):

$$W_k(\mu,\nu) = \left( \int_0^1 d(F_\mu^{-1}(z), F_\nu^{-1}(z))^k dz \right)^{\frac{1}{k}}, \tag{2}$$

where $F_\mu^{-1}$ and $F_\nu^{-1}$ are inverse cumulative distribution functions (CDFs) of $\mu$ and $\nu$, respectively.

**Radon transform and generalized Radon transform:** The Radon transform (Radon, 1917) maps a function $f(\cdot) \in L^1(\mathbb{R}^d)$ to the space of functions defined over spaces of lines in $\mathbb{R}^d$. The Radon transform of $f(\cdot)$ is defined by line integrals of $f(\cdot)$ along all possible hyperplanes in $\mathbb{R}^d$:

$$\mathcal{R}f(t,\theta) = \int_{\mathbb{R}^d} f(x)\delta(t - \langle x,\theta \rangle)dx, \tag{3}$$

where $t \in \mathbb{R}$ and $\theta \in \mathbb{S}^{d-1}$ represent the parameters of hyperplanes in $\mathbb{R}^d$, $\delta(\cdot)$ is the Dirac delta function, and $\langle \cdot,\cdot \rangle$ refers to the Euclidean inner product.

By replacing the inner product $\langle x,\theta \rangle$ in Equation (3) with $\beta(x,\theta)$, a specific family of functions named as *defining function* in Kolouri et al. (2019a), the generalized Radon transform (GRT) (Beylkin, 1984) is defined as:

$$\mathcal{G}f(t,\theta) = \int_{\mathbb{R}^d} f(x)\delta(t - \beta(x,\theta))dx, \tag{4}$$

where $t \in \mathbb{R}$, $\theta \in \Omega_\theta$ while $\Omega_\theta$ is a compact set of all feasible $\theta$, e.g. $\Omega_\theta = \mathbb{S}^{d-1}$ for $\beta(x,\theta) = \langle x,\theta \rangle$ (Kolouri et al., 2019a).

In practice, we can empirically approximate the Radon transform and the GRT of a probability density function $p_\mu$ via:

$$\mathcal{R}p_\mu(t,\theta) \approx \frac{1}{N} \sum_{n=1}^N \delta(t - \langle x_n,\theta \rangle), \tag{5}$$

$$\mathcal{G}p_\mu(t,\theta) \approx \frac{1}{N} \sum_{n=1}^N \delta(t - \beta(x_n,\theta)), \tag{6}$$

where $x_n \sim p_\mu$ and $N$ is the number of samples. Notably, the Radon transform is a linear bijection (Helgason, 1980), and the GRT is a bijection if the defining function $\beta$ satisfies certain conditions (Beylkin, 1984).

**Sliced Wasserstein distance and generalized sliced Wasserstein distance:** By applying the Radon transform to $p_\mu$ and $p_\nu$ to obtain multiple projections, the sliced Wasserstein distance (SWD) decomposes the high-dimensional Wasserstein distance into multiple one-dimensional Wasserstein distances which can be efficiently evaluated (Bonneel et al., 2015). The $k$-SWD between $\mu$ and $\nu$ is defined by:

$$\text{SWD}_k(\mu,\nu) = \left( \int_{\mathbb{S}^{d-1}} W_k^k \big( \mathcal{R}p_\mu(\cdot,\theta), \mathcal{R}p_\nu(\cdot,\theta) \big) d\theta \right)^{\frac{1}{k}}, \tag{7}$$

where the Radon transform $\mathcal{R}$ defined by Equation (3) is adopted as the measure push-forward operator. The GSWD generalizes the idea of SWD by projecting distributions onto hypersurfaces rather than hyperplanes (Kolouri et al., 2019a). The GSWD is defined as:

$$\text{GSWD}_k(\mu,\nu) = \left( \int_{\Omega_\theta} W_k^k \big( \mathcal{G}p_\mu(\cdot,\theta), \mathcal{G}p_\nu(\cdot,\theta) \big) d\theta \right)^{\frac{1}{k}}, \tag{8}$$

where the GRT $\mathcal{G}$ is used as the measure push-forward operator. The Wasserstein distances between one-dimensional distributions can be obtained by sorting projected samples and calculating the distance between sorted samples (Kolouri et al., 2019b): with $L$ random projections, the SWD and GSWD between $\mu$ and $\nu$ can be approximated by:

$$\text{SWD}_k(\mu,\nu) \approx \left( \frac{1}{L} \sum_{l=1}^{L} \sum_{n=1}^{N} |\langle x_{I_x^l[n]}, \theta_l \rangle - \langle y_{I_y^l[n]}, \theta_l \rangle|^k \right)^{\frac{1}{k}}, \tag{9}$$

$$\text{GSWD}_k(\mu,\nu) \approx \left( \frac{1}{L} \sum_{l=1}^{L} \sum_{n=1}^{N} |\beta(x_{I_x^l[n]}, \theta_l) - \beta(y_{I_y^l[n]}, \theta_l)|^k \right)^{\frac{1}{k}}, \tag{10}$$

where $I_x^l$ and $I_y^l$ are sequences consist of the indices of sorted samples which satisfy $\langle x_{I_x^l[n]}, \theta_l \rangle \leq \langle x_{I_x^l[n+1]}, \theta_l \rangle$, $\langle y_{I_y^l[n]}, \theta_l \rangle \leq \langle y_{I_y^l[n+1]}, \theta_l \rangle$ in the SWD, and $\beta(x_{I_x^l[n]}, \theta_l) \leq \beta(x_{I_x^l[n+1]}, \theta_l)$, $\beta(y_{I_y^l[n]}, \theta_l) \leq \beta(y_{I_y^l[n+1]}, \theta_l)$ in the GSWD. It is proved in Bonnotte (2013) that the SWD is a valid distance metric. The GSWD is a valid metric except for its neural network variant (Kolouri et al., 2019a).

## 3 AUGMENTED SLICED WASSERSTEIN DISTANCES

In this section, we propose a new distance metric called the augmented sliced Wasserstein distance (ASWD), which embeds flexible nonlinear projections in its construction. We also provide an implementation recipe for the ASWD.

### 3.1 SPATIAL RADON TRANSFORM AND AUGMENTED SLICED WASSERSTEIN DISTANCE

In the definitions of the SWD and GSWD, the Radon transform (Radon, 1917) and the generalized Radon transform (GRT) (Beylkin, 1984) are used as the push-forward operator for projecting distributions to a one-dimensional space. However, it is not straightforward to design defining functions $\beta(x,\theta)$ (Kolouri et al., 2019a) for the GRT due to certain non-trivial requirements for the function (Beylkin, 1984). In practice, the assumption of the transform can be relaxed, as Theorem 1 shows that as long as the transform is injective, the corresponding ASWD metric is a valid distance metric.

To help us define the augmented sliced Wasserstein distance, we first introduce the *spatial Radon transform* which includes the vanilla Radon transform and the polynomial GRT as special cases (See Remark 2).

**Definition 1.** *Given an injective mapping $g(\cdot): \mathbb{R}^d \to \mathbb{R}^{d_\theta}$ and a probability measure $\mu \in P(\mathbb{R}^d)$ which probability density function (PDF) is $p_\mu$, the spatial Radon transform of $p_\mu$ is defined as*

$$\mathcal{H}p_\mu(t,\theta;g) = \int_{\mathbb{R}^d} p_\mu(x)\delta(t - \langle g(x),\theta \rangle)dx, \tag{11}$$

*where $t \in \mathbb{R}$ and $\theta \in \mathbb{S}^{d_\theta - 1}$ are the parameters of hypersurfaces in $\mathbb{R}^d$.*

**Remark 1.** *Note that the spatial Radon transform can be interpreted as applying the vanilla Radon transform to the PDF of $\hat{x} = g(x)$, where $x \sim p_\mu$. Denote the PDF of $\hat{x}$ by $p_{\hat{\mu}_g}$, the spatial Radon transform defined by Equation (11) can be rewritten as:*

$$
\begin{aligned}
\mathcal{H}p_\mu(t,\theta;g) &= E_{x \sim p_\mu}[\delta(t - \langle g(x),\theta \rangle)], \\
&= E_{\hat{x} \sim p_{\hat{\mu}_g}}[\delta(t - \langle \hat{x},\theta \rangle)] \\
&= \int p_{\hat{\mu}_g}(\hat{x})\delta(t - \langle \hat{x},\theta \rangle)d\hat{x} \\
&= \mathcal{R}p_{\hat{\mu}_g}(t,\theta).
\end{aligned}
\tag{12}
$$

*Hence the spatial Radon transform inherits the theoretical properties of the Radon transform subject to certain conditions of $g(\cdot)$ and incorporates nonlinear projections through $g(\cdot)$.*

In what follows, we use $f_1 \equiv f_2$ to denote functions $f_1(\cdot) : X \to \mathbb{R}$ and $f_2(\cdot) : X \to \mathbb{R}$ that satisfy $f_1(x) = f_2(x)$ for almost $\forall x \in X$.

**Lemma 1.** *Given an injective mapping $g(\cdot) : \mathbb{R}^d \to \mathbb{R}^{d_\theta}$ and two probability measures $\mu,\nu \in P(\mathbb{R}^d)$ whose probability density functions are $p_\mu$ and $p_\nu$, respectively, for all $t \in \mathbb{R}$ and $\theta \in \mathbb{S}^{d_\theta-1}$, $\mathcal{H}p_\mu(t,\theta;g) \equiv \mathcal{H}p_\nu(t,\theta;g)$ if and only if $p_\mu \equiv p_\nu$, i.e. the spatial Radon transform is injective. Moreover, the spatial Radon transform is injective if and only if the mapping $g(\cdot)$ is an injection.*

See Appendix A for the proof of Lemma 1.

**Remark 2.** *The spatial Radon transform degenerates to the vanilla Radon transform when the mapping $g(\cdot)$ is an identity mapping. When $g(\cdot)$ is a homogeneous polynomial function with odd degrees, the spatial Radon transform is equivalent to the polynomial GRT (Ehrenpreis, 2003).*

Appendix B provides the proof of Remark 2.

We now introduce the augmented sliced Wasserstein distance, by utilizing the spatial Radon transform as the measure push-forward operator:

**Definition 2.** *Given two probability measures $\mu,\nu \in P_k(\mathbb{R}^d)$, whose probability density functions are $p_\mu$ and $p_\nu$, respectively, and an injective mapping $g(\cdot) : \mathbb{R}^d \to \mathbb{R}^{d_\theta}$, the augmented sliced Wasserstein distance (ASWD) of order $k \in [1,+\infty)$ is defined as:*

$$
\mathrm{ASWD}_k(\mu,\nu;g) = \left( \int_{\mathbb{S}^{d_\theta-1}} W_k^k\big(\mathcal{H}p_\mu(\cdot,\theta;g),\mathcal{H}p_\nu(\cdot,\theta;g)\big)d\theta \right)^{\frac{1}{k}},
\tag{13}
$$

*where $\theta \in \mathbb{S}^{d_\theta-1}$, $W_k$ is the $k$-Wasserstein distance defined by Equation (1), and $\mathcal{H}$ refers to the spatial Radon transform defined by Equation (11).*

**Remark 3.** *Following the connection between the spatial Radon transform and the vanilla Radon transform as shown in Equation (12), the ASWD can be rewritten as:*

$$
\begin{aligned}
\mathrm{ASWD}_k(\mu,\nu;g) &= \left( \int_{\mathbb{S}^{d_\theta-1}} W_k^k\big(\mathcal{R}p_{\hat{\mu}_g}(\cdot,\theta),\mathcal{R}p_{\hat{\nu}_g}(\cdot,\theta)\big)d\theta \right)^{\frac{1}{k}} \\
&= \mathrm{SWD}_k(\hat{\mu}_g,\hat{\nu}_g),
\end{aligned}
\tag{14}
$$

*where $\hat{\mu}_g$ and $\hat{\nu}_g$ are probability measures on $\mathbb{R}^{d_\theta}$ which satisfy $g(x) \sim \hat{\mu}_g$ for $x \sim \mu$ and $g(y) \sim \hat{\nu}_g$ for $y \sim \nu$.*

**Theorem 1.** *The augmented sliced Wasserstein distance (ASWD) of order $k \in [1,+\infty)$ defined by Equation (13) with a mapping $g(\cdot) : \mathbb{R}^d \to \mathbb{R}^{d_\theta}$ is a metric on $P_k(\mathbb{R}^d)$ if and only if $g(\cdot)$ is injective.*

Proof of Theorem 1 is provided in Appendix C.

## 3.2 NUMERICAL IMPLEMENTATION

We discuss in this section how to realize injective mapping $g(\cdot)$ with *neural networks* due to their expressiveness and optimize it with gradient based methods.

**Injective neural networks:** As stated in Lemma 1 and Theorem 1, the injectivity of $g(\cdot)$ is the sufficient and necessary condition for the ASWD being a valid metric. Thus we need specific architecture designs on

implementing $g(\cdot)$ by neural networks. One option is the family of invertible neural networks (Behrmann et al., 2019; Karami et al., 2019; Song et al., 2019), which are both injective and surjective. However, the running cost of those models is usually much higher than that of vanilla neural networks. We propose an alternative approach by concatenating the input $x$ of an arbitrary neural network to its output $\phi_\omega(x)$:

$$g_\omega(x) = [x, \phi_\omega(x)]. \tag{15}$$

It is trivial to show that $g_\omega(x)$ is injective, since different inputs will lead to different outputs. Although embarrassingly simple, this idea of concatenating the input and output of neural networks has found success in preserving information with dense blocks in the DenseNet (Huang et al., 2017), where the input of each layer is injective to the output of all preceding layers.

**Optimization objective:** We aim to project samples to maximally discriminating hypersurfaces between two distributions, so that the projected samples between distributions are most dissimilar subject to certain constraints on the hypersurface, as shown in Figure 1. Similar ideas have been employed to identify important projection directions (Deshpande et al., 2019; Kolouri et al., 2019a; Paty & Cuturi, 2019) or a discriminative ground metric (Salimans et al., 2018) in optimal transport metrics. For the ASWD, the parameterized injective neural network $g_\omega(\cdot)$ is optimized by maximizing the following objective:

$$\mathcal{L}(\mu, \nu; g_\omega, \lambda) = \left( \int_{\mathbb{S}^{d_\theta - 1}} W_k^k \big( \mathcal{H} p_\mu(\cdot, \theta; g_\omega), \mathcal{H} p_\nu(\cdot, \theta; g_\omega) \big) d\theta \right)^{\frac{1}{k}} - L_\lambda, \tag{16}$$

where $\lambda > 0$ and the regularization term $L_\lambda = \lambda \mathbb{E}_{x,y \sim \mu, \nu} \big[ (||g_\omega(x)||_2 + ||g_\omega(y)||_2) \big]$ is used to control the norm of the output of $g_\omega(\cdot)$, otherwise the projections may be arbitrarily large.

**Remark 4.** *The regularization coefficient $\lambda$ adjusts the introduced non-linearity in the evaluation of the ASWD by controlling the norm of $\phi_\omega(\cdot)$ in Equation (15). In particular, when $\lambda \to \infty$, the nonlinear term $\phi_\omega(\cdot)$ shrinks to $0$. The rank of the augmented space is hence explicitly controlled by the flexible choice of $\phi_\omega(\cdot)$ and implicitly regularized by $L_\lambda$.*

By plugging the optimized $g_{\omega,\lambda}^*(\cdot) = \operatorname*{argmax}_{g_\omega}(\mathcal{L}(\mu, \nu; g_\omega, \lambda))$ into Equation (13), we obtain the empirical version of the ASWD. Notably, the regularization term $L_\lambda$ is only used when maximizing the objective in Equation (16), once the optimization is completed, $L_\lambda$ is not used in the calculation of the ASWD defined by Equation (13). Pseudocode is provided in Appendix D.

## 4 EXPERIMENTS

In this section, we describe the experiments that we have conducted to evaluate performance of the proposed distance metric. The GSWD leads to the best performance in a sliced Wasserstein flow problem reported in Kolouri et al. (2019a) and the DSWD outperforms the compared methods in the generative modeling task examined in Nguyen et al. (2020). Hence we compare performance of the ASWD with the state-of-the-art distance metrics in the same examples and report results as below[1].

To examine the robustness of the ASWD, throughout the experiments, we adopt the injective network architecture given in Equation (15) and set $\phi_\omega$ to be a one layer fully-connected neural network whose outputs' dimension equals its inputs' dimension, with a ReLU layer as its activation function.

### 4.1 SLICED WASSERSTEIN FLOWS

We first consider the problem of evolving a source distribution $\mu$ to a target distribution $\nu$ by minimizing Wasserstein distances between $\mu$ and $\nu$ in the sliced Wasserstein flow task reported in Kolouri et al. (2019a).

$$\partial_t \mu_t = -\nabla \mathrm{SWD}(\mu_t, \nu), \tag{17}$$

where $\mu_t$ refers to the updated source distribution at each iteration $t$. The SWD in Equation (17) can be replaced by other sliced-Wasserstein distances to be evaluated. As in Kolouri et al. (2019a), the *2-Wasserstein distance* was used as the metric for evaluating performance of different distance metrics in this task. The set of hyperparameter values used in this experiment can be found in Appendix E.1.

---

[1]Code to reproduce experiment results is available at : `https://bit.ly/2Y23wOz`.

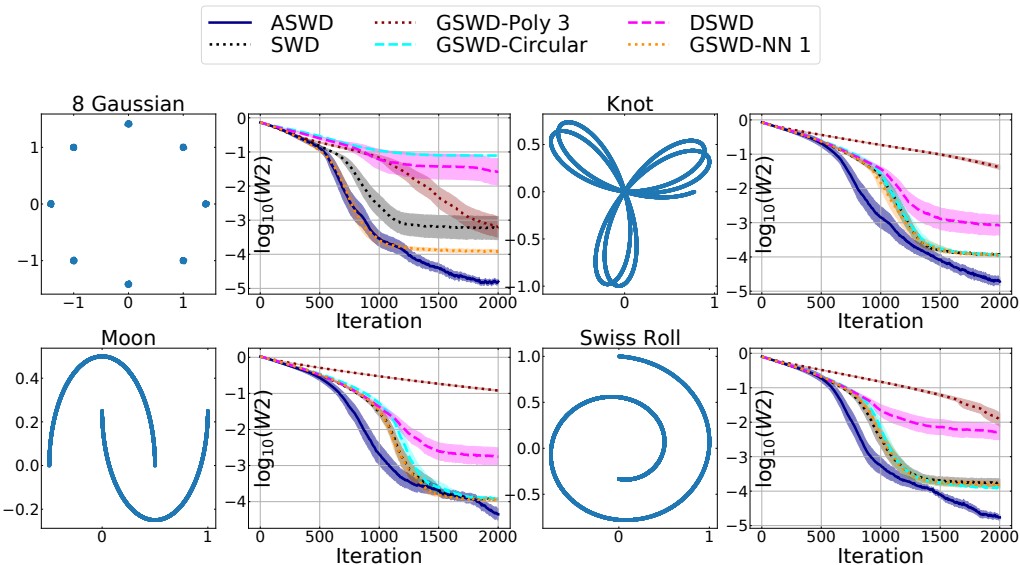

Figure 2: The first and third columns are target distributions. The second and fourth columns are log 2-Wasserstein distances between the target distribution and the source distribution. The horizontal axis show the number of training iterations. Solid lines and shaded areas represent the average values and 95% confidence intervals of log 2-Wasserstein distances over 50 runs. A more extensive set of experimental results can be found in Appendix F.1.

Without loss of generality, we initialize $\mu_0$ to be the standard normal distribution $\mathcal{N}(0,I)$. We repeat each experiment 50 times and record the 2-Wasserstein distance between $\mu$ and $\nu$ at every iteration. In Figure 2, we plot the 2-Wasserstein distances between the source and target distributions as a function of the training epochs and the 8-Gaussian, the Knot, the Moon, and the Swiss roll distributions are respective target distributions. For clarity, Figure 2 displays the experiment results from the 6 best performing distance metrics, including the ASWD, the DSWD, the SWD, the GSWD-NN 1, which directly generates projections through a one layer MLP, as well as the GSWD with the polynomial of degree 3, circular defining functions, out of the 12 distance metrics we compared.

We observe from Figure 2 that the ASWD not only leads to lower 2-Wasserstein distances, but also converges faster by achieving better results with fewer iterations than the other methods in these four target distributions. A complete set of experimental results with 12 compared distance metrics and 8 target distributions are included in Appendix F.1. The ASWD outperforms the compared state-of-the-art sliced-based Wasserstein distance metrics with 7 out of the 8 target distributions except for the 25-Gaussian. This is achieved through the simple injective network architecture given in Equation (15) and a one layer fully-connected neural network with equal input and output dimensions throughout the experiments. In addition, ablation study is conducted to study the effect of injective mappings, the regularization coefficient $\lambda$, and the optimization of hypersurfaces in the ASWD. Details can be found in Appendix F.2.

## 4.2 GENERATIVE MODELING

In this experiment, we use the sliced-based Wasserstein distances for a generative modeling task described in Nguyen et al. (2020). The task is to generate images using generative adversarial networks (GANs) (Goodfellow et al., 2014) trained on either the CIFAR10 dataset (64×64 resolution) (Krizhevsky, 2009) or the CELEBA dataset (64×64 resolution) (Liu et al., 2015). Denote the hidden layer and the output layer of the discriminator by $h_\psi$ and $D_\Psi$, and the generator by $G_\Phi$, we train GAN models with the following objectives:

$$\min_\Phi \text{SWD}(h_\psi(p_r), h_\psi(G_\Phi(p_z))), \tag{18}$$

$$\max_{\Psi,\psi} \mathbb{E}_{x \sim p_r}[\log(D_\Psi(h_\psi(x)))] + \mathbb{E}_{z \sim p_z}[\log(1 - D_\Psi(h_\psi(G_\Phi(z))))], \tag{19}$$

Table 1: FID scores of generative models trained with different distance metrics. Lower scores indicate better image qualities. $L$ is the number of projections, we run each experiment 10 times and report the average values and standard errors of FID scores for CIFAR10 dataset and CELEBA dataset. The running time per training iteration for one batch containing 512 samples is computed based on a computer with an Intel (R) Xeon (R) Gold 5218 CPU 2.3 GHz and 16GB of RAM, and a RTX 6000 graphic card with 22GB memories.

| $L$ | SWD | | GSWD | | DSWD | | ASWD | |
|---|---|---|---|---|---|---|---|---|
| | FID | $t$ (s/it) | FID | $t$ (s/it) | FID | $t$ (s/it) | FID | $t$ (s/it) |
| CIFAR10 | | | | | | | | |
| 10 | 192.6±5.7 | 0.32 | 189.5±6.0 | 0.35 | 79.0±4.2 | 0.48 | **73.2±3.1** | 0.55 |
| 100 | 155.0±2.9 | 0.32 | 155.9±3.2 | 0.70 | 72.2±8.2 | 0.51 | **66.7±3.2** | 0.57 |
| 1000 | 126.0±2.9 | 0.34 | 134.5±2.7 | 2.10 | 74.3±4.3 | 1.22 | **65.5±3.9** | 1.32 |
| CELEBA | | | | | | | | |
| 10 | 118.3±3.1 | 0.32 | 143.2±5.5 | 0.35 | 105.3±3.4 | 0.49 | **99.2±4.3** | 0.53 |
| 100 | 116.0±2.8 | 0.33 | 120.8±1.8 | 0.69 | 103.1±3.8 | 0.51 | **94.3±2.2** | 0.56 |
| 1000 | 104.4±2.8 | 0.34 | 101.8±1.8 | 2.14 | 97.4±2.1 | 1.21 | **90.5±3.0** | 1.31 |

(a) FID scores on CIFAR10 ($L=1000$)    (b) FID scores on CELEBA ($L=1000$)

Figure 3: FID scores of generative models trained with different metrics on CIFAR10 and CELEBA datasets with $L=1000$ projections. The error bar represents the standard deviation of the FID scores at the specified training epoch among 10 simulation runs.

where $p_z$ is the prior of latent variable $z$ and $p_r$ is the distribution of real data. The SWD in Equation (18) is replaced by the ASWD and other variants of the SWD to compare their performance. The GSWD with the polynomial defining function and the DGSWD is not included in this experiment due to its excessively high computational cost in high-dimensional space.The *Fréchet Inception Distance* (FID score) (Heusel et al., 2017) is used to assess the quality of generated images. More details on the network structures and the parameter setup used in this experiment are available in Appendix E.2.

We run 200 and 100 training epochs to train the GAN models on the CIFAR10 and the CELEBA dataset, respectively. Each experiment is repeated for 10 times. We report experimental results in Table 1. With the same number of projections and a similar computation cost, the ASWD leads to significantly improved FID scores among all evaluated distances metrics on both datasets, which implies that images generated with the ASWD are of higher qualities. Figure 3 plots the FID scores recorded during the training process. The GAN model trained with the ASWD exhibits a faster convergence as it reaches lower FID scores with fewer epochs. Randomly selected samples of generated images are presented in Appendix G.

## 5    CONCLUSION

We proposed a novel variant of the sliced Wasserstein distance, namely the augmented sliced Wasserstein distance (ASWD), which is flexible, has high projection efficiency, and generalizes well. The ASWD adaptively updates the hypersurface where the samples are projected onto by learning from data. We proved that the ASWD is a valid distance metric and presented its numerical implementation. We

reported empirical performance of the ASWD over state-of-the-art sliced Wasserstein metrics in numerical experiments. The ASWD leads to the smallest distance errors over the majority of datasets in a sliced Wasserstein flow task and superior performance in a generative modeling task.

The ASWD can be extended in several directions. One future research topic is to incorporate the framework of the ASWD into the sliced Gromov-Wasserstein distance, such that data-adaptive, nonlinear projections can be learned to compare distributions whose supports do not necessary lie in the same metric space. It would be also interesting to explore the identification of the barycenter of objects via the ASWD, e.g. through an optimization scheme alternating between optimization of the nonlinear maps in the ASWD framework and the barycenter. Another interesting area is the incorporation of the ASWD with state-of-the-art generative models to further exploit projection efficiency. We anticipate that they will provide promising directions for enabling the generalization for sliced Wasserstein distance to be applied in wider domains.

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
