# OpenReview forum: "Augmented Sliced Wasserstein Distances"
_ICLR.cc/2021/Conference — Reject_

### Official Review · AnonReviewer2 · 2020-10-26
**The metrics is hard to interpret**

**Rating:** 4
**Confidence:** 3

**Review:**

Augmented Sliced Wasserstein distance between random vectors X&Y is defined as SWD between g(X) and g(Y) for an injective g.

It is claimed that this generalization solves the problem with SWD, i.e. a small distance for most of the projection directions.  Since the paper is purely experimental, then it is better to demonstrate that this issue is resolved (it is not obvious why the problem disappears).

Since g_omega is also searched for by maximizing (16), then it becomes hard to interpret the value of metrics (also taking into account the role of lambda). What it measures now?

The code is given, results seem reproducible. Reported results with generative modeling compare ASWD with SWD, GSWD, DSWD. Since the paper is experimental, maybe it is natural to expect a comparison with SOTA generators that do not deal with generalizations of Wasserstein distance. Also, fake images generated by SWD, GSWD, DSWD are not given (only FID is shown).

---

> ### Author Response · Authors · 2020-11-18
> **Author response to AnonReviewer 2**
>
> (1) "*It is claimed that this generalization solves the problem with SWD, i.e. a small distance for most of the projection directions. Since the paper is purely experimental, then it is better to demonstrate that this issue is resolved (it is not obvious why the problem disappears).*"
>
> We first would like to thank the reviewer for the review. We respectfully disagree that the paper is purely experimental - we developed a novel technique, characterized its theoretical properties through one theorem and two lemmas as well as their proofs, presented the numerical implementation recipe, and provided the results of numerical experiments that indicate it significantly outperforms state-of-the-art slicing-based OT metrics in the same experiments that those metrics were originally evaluated. Hence, the contributions of this paper are on the methodology side validated by both theoretical properties and experiment results.
>
> The issue of low projection efficiency in the SWD is addressed through optimizing a hypersurface where the compared distributions are projected onto, such that the projected samples are well-separated and the projection efficiency is improved. Hence, the proposed ASWD is well-motivated and we demonstrated that the ASWD is a valid metric with injective mapping (Theorem 1). Figure 1 serves an intuitive example to explain the high projection efficiency of the ASWD and in Section 4 we presented superior performance of the ASWD.
>
> (2) "*Since $g_\omega$ is also searched for by maximizing (16), then it becomes hard to interpret the value of metrics (also taking into account the role of $\lambda$). What it measures now?*"
>
> Like the definitions of the SWD and the GSWD, the ASWD can be interpreted as a metric that measures the average Wasserstein distances between projected one-dimensional distributions. The difference is that previous work projects samples via a specific defining function, while the defining function of the ASWD can be learned from data.
>
> The regularization coefficient $\lambda$ is only used when optimizing the hypersurface, once the optimization is completed, $\lambda$ has no effect on the calculation of the ASWD. In particular, $\lambda$ is not involved in the ASWD defined by Eq. (13) if the mapping $g_\omega(\cdot)$ has been learned. We have clarified this in the last paragraph of Section 3 in the updated paper to remove the ambiguity.
>
> (3) "*Since the paper is experimental, maybe it is natural to expect a comparison with SOTA generators that do not deal with generalizations of Wasserstein distance. Also, fake images generated by SWD, GSWD, DSWD are not given (only FID is shown).*"
>
> As in our comment to (1), we disagree that this paper is experimental. The purpose of the included generative modelling experiment is to compare the proposed method with other state-of-the-art sliced-based Wasserstein metrics (DSWD in this example) in the same experiment setup which the DSWD has been evaluated and reached favorable performance. Comparison with other generators that are unrelated with sliced-based Wasserstein metrics is a topic worth further investigation, but we feel that this is out of the scope of this paper and has included as future research topic in Section 5 of the revised version of the paper. Fake images generated by SWD, GSWD, and DSWD have been added in Appendix G.

---

### Official Review · AnonReviewer3 · 2020-10-27
**Valuable contributions**

**Rating:** 7
**Confidence:** 3

**Review:**

This paper introduces the augmented sliced Wasserstein distances (ASWD) to capture non-linear projections, as opposed to sliced Wasserstein distances. To this end, the architecture first maps inputs to higher dimensional hypersurfaces, before projecting. The designed ASWD is shown to be a metric as long as the initial mapping is injective.

Numerically, the initial mapping is parameterized as the concatenation of the input and its output using a neural network. The loss aims at capturing a hypersurface that differentiates measures the most, and is regularized to control the initial mapping. The first task considers the minimization of sliced Wasserstein flows for different settings of synthetic data. The other task consists of generating images with GANs, using a sliced Wasserstein loss to learn the generator network.

Strong points of the paper include:
1.	The paper is well written. The literature review is very thorough and comparison with the proposed method is well explained.
2.	Both theoretically and numerically, the method is well described, well compared to existing notions and yields convincing results.

Drawbacks / questions:
1.	It seems that a lot of projections are needed to retrieve visually satisfactory generated examples.
2.	Have you tried other types of injective maps than Eq 15?
3.	Can you comment on the impact of regularization strength lambda in practice?

I recommend an accept for this paper, which, to the best of my knowledge, brings both theoretical and numerical valuable contributions to the literature.

---

> ### Author Response · Authors · 2020-11-18
> **Author response to AnonReviewer 3**
>
> (1) "*It seems that a lot of projections are needed to retrieve visually satisfactory generated examples.*"
>
> We would like to thank the reviewer for the feedback on our paper. We acknowledge that 1000 projections are beneficial in retrieving visually satisfactory generative modelling experiment of our paper, by slicing through 512-dimensional distributions in the particular framework used in both [1] and [2], two of the recent work in sliced-based Wasserstein metrics. We have shown that the ASWD with 100 projections provides significantly smaller FID scores compared with SWD, GSWD and DSWD with 1000 projections. We have now also included the images generated by those methods in Appendix G. By utilizing a different generative model other than the one used in [1], we may achieve similarly visually satisfactory examples with ASWD using less projections. But we consider this as out of scope for this paper so we include it as a future research direction in Section 5 of the revised paper, as the goal here is to compare across slicing-based optimal transport metrics in the same setup that the state-of-the-art metrics were originally evaluated.
>
> (2) "*Have you tried other types of injective maps than Eq 15?*"
>
> We have now included in Appendix F.2 results from the planar flow and radial flow [3], two variants of invertible mappings, as alternatives to the injective mapping specified by Equation (15). From the numerical results presented in Figure 7, we found that the ASWD defined with planar flow and radial flow produced better performance than GSWD variants in most setups. They exhibit slightly worse performance compared with the ASWD with injective mapping defined in Equation (15), possibly due to the additional restriction in invertible mapping imposed by the planar flow and radial flow.
>
> (3) "*Can you comment on the impact of regularization strength lambda in practice?*"
>
> In the sliced Wasserstein flow experiment, the value of $\lambda$ is selected from a candidate set of \{0.01, 0.05, 0.1, 0.5\}, we found the empirical performance is not sensitive to the choice of $\lambda$ in that candidate set. The comparison between ASWDs trained with different values of $\lambda$ has been added in the Appendix F.2 (Figure 6).
>
> [1] K. Nguyen, N. Ho, T. Pham, and H. Bui. Distributional sliced-Wasserstein and applications to generative modeling. *arXiv preprint arXiv:2002.07367*, 2020.
>
> [2] I. Deshpande, Z. Zhang, and A. G. Schwing. Generative modeling using the sliced Wasserstein distance. In *Proc. IEEE conference on Computer Vision and Pattern Recognition (CVPR)*, pages 3483–3491, Salt Lake City, Utah, USA, 2018.
>
> [3] D. Rezende and S. Mohamed. Variational inference with normalizing flows. In *International Conference on Machine Learning (ICML)*, pages 1530–1538, Lille, France, 2015.

---

### Official Review · AnonReviewer1 · 2020-10-30
**Clarification of numerical results necessary**

**Rating:** 6
**Confidence:** 4

**Review:**

The paper provides a notion of generalisation for sliced Wasserstein distances, that allows to explore nonlinear projections in arbitrary subspaces in a suitable and efficient way. The paper is well-written and provides a novel solution to the problem of exploiting nonlinear subspaces in computing distances. I feel this idea, although simple can be quite powerful, and can be extended to wide domains especially identifying objects based on arbitrary feature selections although I do not think the authors have explored that direction in this piece of work.
Comments:
(1) Numerical results show the distance computed by ASWD is smaller than other measures. This can, however, be misleading in the sense that it might also be obtained by insufficient exploration of non-linear subspaces. For eg, if the sliced Wasserstein distances are computed along orthogonal directions to the primary features the Wasserstein distance obtained in that regard will also be very small, although this does not in anyway validate the superiority of the distance metric. I feel this argument therefore needs some clarification.
(2) Can I use this method to construct arbitrary nonlinear projections of choice instead of the projection that gives the best distinction? For eg, suppose two pictures have several objects among which cars in the two pictures are the most distinguishing features. However, I also want to see what other features can help distinguish these two pictures from other pictures which do not have cars. How do I do that? In that broader sense I suppose my question deals with trying to identify barycenters of objects via sliced Wasserstein distances. Any thoughts in that regard could be useful for the reader.

Overall I find the paper an interesting read although it requires some clarifications as outlined above.

---

> ### Author Response · Authors · 2020-11-18
> **Author response to AnonReviewer 1**
>
> (1) "*Numerical results show the distance computed by ASWD is smaller than other measures. This can, however, be misleading in the sense that it might also be obtained by insufficient exploration of non-linear subspaces.* "
>
> We thank the reviewer for the comments. We have clarified in Section 4.1 of the paper that ''As in Kolouri et al. (2019a), the *2-Wasserstein distance* was used as the metric for evaluating performance of different distance metrics in this task''. We would like to clarify that in this sliced Wasserstein flow experiment, the numerical results presented in Figure 2 show that the ASWD produces lower 2-Wasserstein distances, which are exact Wasserstein distances instead of ASWDs, as a result of better sliced Wasserstein flow due to the incorporation of the ASWD. But this does not imply that the ASWD is smaller than the other measures. On the contrary, as we presented in Figure 1, given two distributions, since the ASWD projects samples onto distinguishing hypersurfaces where the projected samples are well-separated, the ASWD is larger than the SWD in expectation. The idea behind the ASWD is that we would like to find hypersurfaces that distinguish between two (empirical) distributions under *random* projections. So, we maximize the objective specified by Equation (16), instead of minimizing it.
>
> When employing the ASWD as the distance metric, small distances between projected samples indeed can be obtained if the distance is calculated along orthogonal directions. However, since multiple random projections are involved and the distances between projected samples are averaged in the calculation of the ASWD, orthogonal projections generally have no significant impact on the value of the ASWD, coupled with the fact that the nonlinear mapping $g(\cdot)$ is learned to maximize the ASWD under random projections.  In fact, Figure 1 shows that ASWD produces much more significant distances under random projections compared to the SWD.
>
> (2) "*Can I use this method to construct arbitrary nonlinear projections of choice instead of the projection that gives the best distinction?  In that broader sense I suppose my question deals with trying to identify barycenters of objects via sliced Wasserstein distances.*"
>
> Surely you can. As you mentioned, in our work, the mapping $g(\cdot)$ is optimized so that random projections of the mapped samples between two distributions give the best distinction. In other applications, one may explore other optimization objectives and schemes. For example, for finding the barycenter of objects via slicing-based Wasserstein distance, there are two potential directions to explore - a straightforward approach is to optimize the mapping $g(\cdot)$ by maximizing the ASWD between objects, then use the learned mapping to calculate the barycenter; another way is an alternating optimization scheme, alternating between optimizing $g(\cdot)$ by maximizing the weighted average ASWD between the barycenter and the objects, and optimizing the barycenter by minimizing this weighted average ASWD distance (the optimization objective specified in Eq. (32) of [1]). This is a very interesting topic but we consider it out of scope for this paper. We have now added this in the paper as a future research direction in Section 5.
>
> [1] N. Bonneel, J. Rabin, G. Peyré, and H. Pfister. Sliced and Radon Wasserstein barycenters of measures. *Journal of Mathematical Imaging and Vision*, 51(1):22–45, 2015.

---

### Author Response · Authors · 2020-11-19
**Summary of changes in revision**

We thank all the reviewers for your valuable feedback. We have updated the paper and appendices according to your suggestions. The main revisions we made are:


(1) (AnonReviewer 1) We have clarified numerical results in the individual response below and in Section 4.1 of the paper. We have included discussions on future research directions in Section 5, including the extension of the ASWD in computing the barycenter.

(2) (AnonReviewer 3) We reported and discussed in Appendix F.2 the numerical results from two additional experiments to investigate the impact from the choice of injective mapping $g(\cdot)$ and regularization coefficient $\lambda$ on the performance of the ASWD.

(3) (AnonReviewer 2) Fake images generated by the SWD, GSWD, and DSWD have been added in Appendix G. We have also added clarification in the last paragraph of Section 3 that the regularization term is only used to optimize the hypersurface, and has no effect on the calculation of the ASWD after the optimization is completed.


We addressed reviewers' individual concerns and questions in the response. We are looking forward to having further discussions with the reviewers if there is anything that requires further clarification.

---

### Decision · Program_Chairs · 2021-01-07
**Final Decision**

**Decision:**

Reject

**Comment:**

Given two data measures in R^d, this paper proposes to use a NN to augment the representation of each data point found in these measures with additional coordinates. The measures are then compared using the sliced Wasserstein distance on these augmented representations. Because this augmentation is injective by design (the original vectors are part of the new representation) simple metric properties are kept. The authors propose to learn in a robust/adversarial way these augmentations. They propose simple experiments to illustrate that idea.

Although I found the idea interesting, I think it falls short of acceptance at ICLR. I agree with the sentiment of other reviewers 1 and 2 that defining another variant of robust/NN inspired variant of the W distance is interesting, but at this point the readership of the conference expects more than simple experiments on toy data and hard to interpret GAN results. I think there is value in the draft as it stands now, but that more efforts are needed to convince this variant is scalable / useful for other downstream tasks (e.g. W barycenters, or other easier to interpret W problems in lower dimensions).

minor comments
- as it stands, equation 2 is wrong if you do not add more conditions on the cost function d(.,.).
- " the idea of SWD by projecting distributions onto hypersurfaces rather than hyperplane" -> this is wrong, the projection is done onto lines or curves, not hyperplanes or hypersurfaces.